# Root zone soil moisture in over 25 % of global land permanently beyond pre-industrial variability as early as 2050 without climate policy

En Ning Lai[1], Lan Wang-Erlandsson[2], Vili Virkki[3], Miina Porkka[3], and Ruud J. van der Ent[1]

[1]Department of Water Management, Faculty of Civil Engineering and Geosciences, Delft University of Technology, Delft, Netherlands
[2]Stockholm Resilience Centre, Stockholm University, Stockholm Sweden
[3]Water and Development Research Group, Aalto University, Espoo, Finland

**Correspondence:** Ruud J. van der Ent (r.j.vanderent@tudelft.nl)

**Abstract.** Root-zone soil moisture is a key variable representing water cycle dynamics that strongly interacts with ecohydrological, atmospheric, and biogeochemical processes. Recently, it was proposed as the control variable for the green water planetary boundary, suggesting that widespread and considerable deviations from baseline variability now predispose Earth System functions critical to an agriculture-based civilisation to destabilization. However, the global extent and severity of root-zone soil moisture changes under future scenarios remains to be scrutinized. Here, we analyzed root-zone soil moisture departures from the pre-industrial climate variability for a multi-model ensemble of 14 Earth System Models (ESMs) in the Coupled Model Intercomparison Project Phase 6 (CMIP6) in four climate scenarios as defined by the Shared Socioeconomic Pathways (SSPs), SSP1-2.6, SSP2-4.5, SSP3-7.0, and SSP5-8.5, between 2021 and 2100. The analyses were done for 43 ice-free climate reference regions used by the Intergovernmental Panel on Climate Change (IPCC). We defined 'permanent departures' when a region's soil moisture exits the regional variability envelope of the pre-industrial climate and does not fall back into the range covered by the baseline envelope until 2100. Permanent dry departures (i.e. lower soil moisture than pre-industrial variability) were found to be most pronounced in Central America, southern Africa, the Mediterranean region, and most of South America, whereas permanent wet departures are most pronounced in southeastern South America, northern Africa, and southern Asia. In the Mediterranean region, dry permanent departure may have already happened according to some models. By 2100, there is dry permanent departures in the Mediterranean in 70 % of the ESMs in SSP1-2.6, the most mitigated situation, and more than 90 % in SSP3-7.0 and SSP5-8.5, the medium-high and worst-case scenarios. Northeastern Africa is projected to experience wet permanent departures in 64 % of the ESMs under SSP1-2.6, and 93 % under SSP5-8.5. The percentage of ice-free land area with departures increases in all SSP scenarios as time goes by. Wet departures are more widespread than dry departures throughout the studied timeframe, except in SSP1-2.6. In most regions, the severity of the departures increases with the severity of global warming. In 2050, permanent departures (ensemble median) occur in about 10 % of global ice-free land areas in SSP1-2.6, and in 25 % in SSP3-7.0. By the end of the 21st century, the occurrence of permanent departures in SSP1-2.6 increases to 34 %, and in SSP3-7.0, 45 %. Our findings underscore the importance of mitigation to avoid further degrading the Earth System functions upheld by soil moisture.

# 1 Introduction

Soil moisture's role in land-atmosphere interactions, ecohydrological regulation, biogeochemical cycles and land surface hydrology makes it a momentous component of the Earth system that is also critically important for food security, agriculture-based economies, and broader societal functions (Webster et al., 1998). The availability of soil moisture in the root-zone critically controls biomass production, transpiration, and carbon and nutrient cycling of terrestrial ecosystems and production systems (Rigden et al., 2020; Kurc and Small, 2007). Soil moisture deficiencies are detrimental to crop production, forest resilience, and land carbon sequestration and storage, whereas excessive soil moisture likewise can lead to water logging, altered regional energy balance, and seasonal or even inter-annual atmospheric circulation (Douville et al., 2001).

Root-zone soil moisture is now substantially modified by human activities, such as greenhouse gas emissions, deforestation, aerosol pollution, freshwater depletion, and their interactions (Chrysafi et al., 2022). These human modifications of root-zone soil moisture and the Earth system implications of its changes prompted the suggestion to include root-zone soil moisture in the planetary boundaries framework as the green water planetary boundary (Wang-Erlandsson et al., 2022). The planetary boundaries framework demarcates a safe operating space for humanity based on the relatively stable Earth system conditions during the Holocene that enables agriculture-based civilisations to thrive (Steffen et al., 2015; Rockström et al., 2009). Wang-Erlandsson et al. (2022) postulated that root zone soil moisture is the most suitable variable to demonstrate the impact of changes in the Earth System on the green water cycle (precipitation, soil moisture and evaporation) as it is most closely related to ecological and climatic shifts. Both drying and wetting are considered, as both are able to impact Earth system functioning. For example, low soil moisture content can cause or contribute to wildfires, heatwaves, and self-amplified forest loss (Zemp et al., 2017); whereas high soil moisture content can be associated with high water levels in wetlands and increased anaerobic methane production, delay of monsoon onset (Moon and Ha, 2019), and, when concurring with warm air, lead to killer heatwaves above wet bulb temperatures (Raymond et al., 2020). Porkka et al. (2023) analyzed the pre-industrial soil moisture variability and the percentage of ice-free global land area in which soil moisture is outside the pre-industrial variability envelope (defined as the 5th and 95th percentile). They found that on average 16 % of the global land area was outside this envelope in 2005 (whereas the expected value would be 10 % in case of no change to the water cycle dynamics). These widespread soil moisture changes, together with evidence of soil-moisture impacts on ecological and climatic shifts, led to the provisional assessment that freshwater changes are no longer within a planetary safe zone (Wang-Erlandsson et al., 2022; Porkka et al., 2023).

Since root zone soil moisture conditions appear to be outside the planetary safe zone at present, it is important to study how root zone soil moisture will change in possible future climates. Dirmeyer et al. (2016) explicitly studied the future hydrological cycle's deviation from the historical conditions based on a Coupled Model Intercomparison Project Phase 5 (CMIP5) ensemble. They studied when soil moisture in the upper 10 cm in two Representative Concentration Pathway (RCP) scenarios, RCP4.5 and RCP8.5, would permanently exceed the historical maximum and go below the minimum, which were determined from the historical simulation (1860-2005). They found that permanently less soil moisture (drying) is expected in many regions across the globe in the boreal summer, while there is a strong tendency for permanently more soil moisture (wetting) during winters

in the high northern latitudes. Another study analyzed drought projections of the 21st century (Cook et al., 2020) CMIP6 data to investigate the differences between the total soil moisture content in a future period (2071-2100) for different scenarios compared to the historical (1850-2014) mean. This study showed that drying in surface soil moisture is more pervasive and extensive than that in the total soil column. Robust decrease in soil moisture content occurs in western North America, Central America, Europe and the Mediterranean, the Amazon, southern Africa, China, Southeast Asia, and Australia especially in the summer. Furthermore, in the Sixth Assessment Report (AR6) Douville et al. (2021) of the Intergovernmental Panel on Climate Change (IPCC) it was concluded that, by the end of the 21st century, soil moisture content will decrease in semi-arid regions, and aridification is expected, with high confidence, to profoundly surpass what was observed in the past millennium in the Mediterranean, Central Chile, and western North America even in low-emissions climate scenarios. However, when and where (permanent) root-zone soil moisture departures from pre-industrial variability - following the recent green water planetary boundary definition (Wang-Erlandsson et al., 2022) - are expected in the future is an outstanding issue.

The goal of this study is to determine the future climatic state of root zone soil moisture for possible future climates (2021-2100). Hereto we use a CMIP6 multi-model ensemble of total soil moisture with the Pre-industrial Control (PiControl) scenario as baseline and four different future scenarios of Shared Socioeconomic Pathways (SSP) combined with emissions scenarios: SSP1-2.6, SSP2-4.5, SSP3-7.0, and SSP5-8.5 (O'Neill et al., 2016; Riahi et al., 2017). We aggregate all data to the scale of the IPCC WGI climate reference regions (Iturbide et al., 2020), which differs from the scale and aggregation used in Porkka et al. (2023), but is justified as this study is more focused on highlighting specific regions rather than obtaining precise global numbers. For all regions we determine departures from the baseline on monthly and yearly scales. Furthermore, we analyze whether the departures are permanent and determine the 'time of emergence', which is the moment that soil moisture moves beyond and does not fall back within the envelope of baseline variability before 2100. Further, we quantify the global land area with soil moisture content departures, which provides scenarios for the future status of the green water planetary boundary. As such, this study could act as early warning for regions that are projected to have a permanently different water cycle in the coming decades.

## 2 Methods

### 2.1 Data

Here, we use total soil moisture content from 14 different ESMs based on CMIP6 output (Eyring et al., 2016). The total soil moisture content is the mass of water in all phases and in all soil layers. Whereas the green water planetary boundary definition is based on root zone soil moisture (Wang-Erlandsson et al., 2022), this was not explicitly available from most ESMs in CMIP6. Depending on the model configuration the total soil moisture may coincide with root zone soil moisture in most areas (e.g., van Oorschot et al., 2021). In any case, we deem it logical to assume that any changes occurring in the total soil moisture in fact are occurring in the hydrological active zone, which is the zone in which plant roots are active (e.g., Feddes et al., 2001; Fan et al., 2017; Singh et al., 2020), and, therefore, we focus on analyzing the absolute and not the relative changes in total soil moisture.

In this study, the total soil moisture content in four different SSPs from ScenarioMIP was compared with the baseline, which is the pre-industrial control (PiControl) simulation that is a part of the Diagnostic, Evaluation and Characterization of Klima (DECK) experiments. The PiControl simulation is done based on non-evolving pre-industrial conditions in which the Earth System is mostly undisturbed by humans. The year 1850 is the reference year for this period (Eyring et al., 2016).

SSPs are scenarios used to generate different radiative forcing pathways by estimating future green house gasses (GHGs) emissions and land-use change scenarios based on integrated assessments and assumptions regarding socio-economic developments, climate mitigation efforts, and global governance (Kriegler et al., 2017). The SSPs used in this study are SSP1-2.6 (+2.6 W m$^{-2}$; low GHGs emissions), SSP2-4.5 (+4.5 W m$^{-2}$; intermediate GHGs emissions), SSP3-7.0 (+7.0 W m$^{-2}$; high GHGs emissions), and SSP5-8.5 (+8.5 W m$^{-2}$; very high GHGs emissions) (Riahi et al., 2017). Note that SSP2-4.5 is roughly on the current pathway of emission reductions and SSP3-7.0 is an average 'no climate policy'-scenario (Hausfather and Peters, 2020). For our data ensemble, we selected all ESMs that provided simulation outputs for both the PiControl as well as all four SSPs of interest on the ESGF servers as of 1 November 2021, which amounted to 14 models. We analyzed the output data from these models as reported on the ESGF servers on 25 June 2022 (ESGF, 2022). Detailed information on these models can be found from the references provided in Table S1.

## 2.2 Study area

The total soil moisture content was studied on a regional scale to better illustrate the effect of climate change on regional hydrological cycles. Although Wang-Erlandsson et al. (2022) and Porkka et al. (2023) used a 0.5° latitude × 0.5° longitude grid scale for the analysis, we chose to adopt the framework of the IPCC WGI climate reference regions, as stated in the Fig. S1 in the supplement (Iturbide et al., 2020), which bypasses the issue of downscaling different climate model data and allows for easier interpretation of the subcontinental analysis of climate model data.

## 2.3 Baseline; Wet and Dry Departures; Time of Emergence

Each ESM's 30-year average regional soil moisture content between 2071 and 2100 from each SSP (the shaded region in the right figure of Fig. 1) was calculated and compared with the 80-year average of the PiControl data (the shaded region in the left figure of Fig. 1). The differences between the two averages were recorded as the regional deviations from the baseline. The ESM ensemble's median regional deviations is shown in Section 3.1.

The total soil moisture content of different SSPs between 2021 and 2100 was compared with the PiControl baseline to determine whether and when the deviation of the total soil moisture content from the baseline becomes permanent. The framework we used is illustrated in Fig. 1 using one particular model and region as an example. The PiControl baseline is enclosed by the 5th and 95th percentile of the PiControl data. Following a similar approach as Dirmeyer et al. (2016), we defined the time of emergence as when the yearly mean total soil moisture content starts to deviate permanently from the PiControl baseline (determined separately for each ESM and region), within the studied timeframe of 2021 to 2100. Permanent wet departure occurs if the yearly mean total soil moisture content exceeds the 95th percentile and does not fall back within the envelope of baseline variability before 2100, while permanent dry departure occurs if the soil moisture content goes below the 5th percentile. The

time of emergence was determined for every climate region except Greenland and Antarctica since they consist mainly of
permanently frozen soils where we did not consider soil moisture to be a meaningful metric in line with Wang-Erlandsson
et al. (2022) and Porkka et al. (2023). If the apparent permanent departure was after 2095, it was not classified as permanent
because we considered 2095 too close to the end of the studied timeframe, and hence there is not enough information to in-
dicate whether the departure is indeed permanent. To visualize the results, the net number of ESMs that project a permanent
departure in each region was computed by subtracting the number of models with a permanent dry departure from the number
of models with a permanent wet departure. The result is shown in Section 3.2.

The monthly regional time of emergence for total soil moisture content was also analyzed. The regional total soil moisture
content data in the SSPs and PiControl scenario were first grouped by month. The regional time of emergence for each month
was determined from the grouped data using the method illustrated in Fig. 1. For each ensemble member, the number of months
with a permanent departure was then calculated for every climate region. The ensemble mean is shown in Section 3.3.

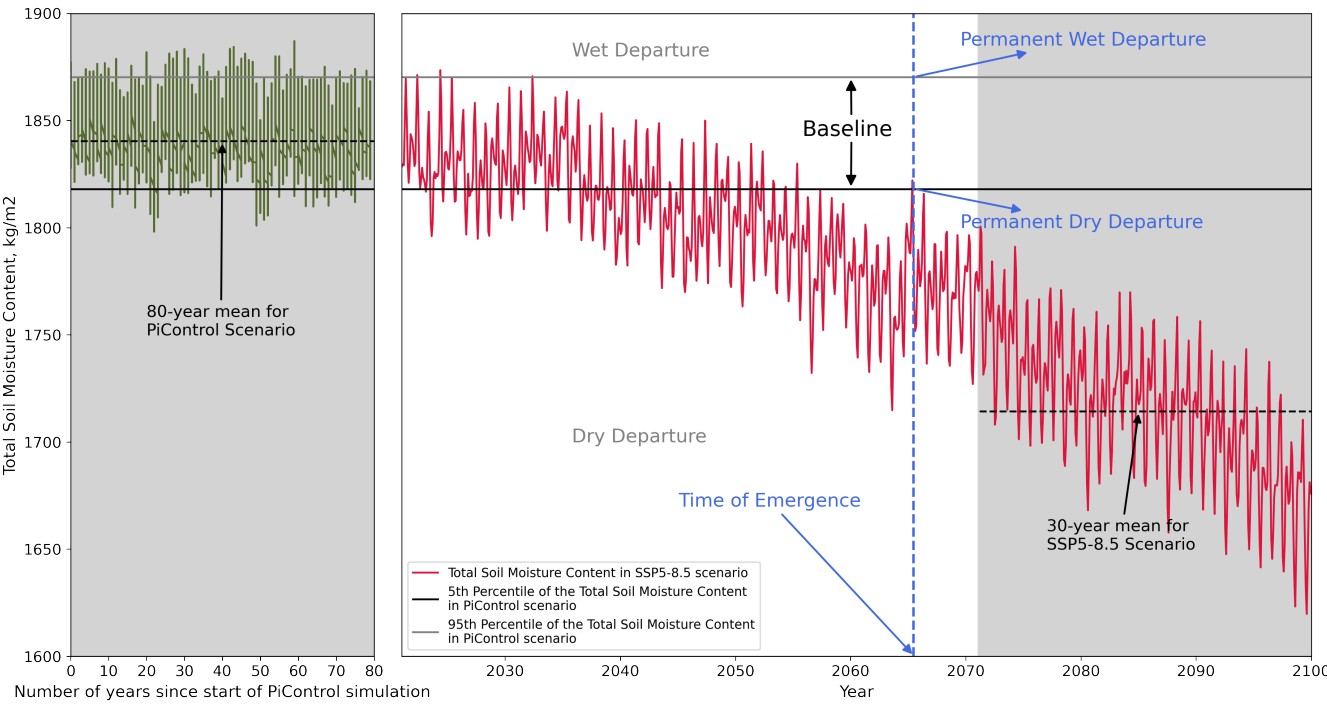

**Figure 1.** Illustration of concepts used in this study. **Left**: determination of the PiControl baseline (climate conditions around 1850). **Right**: determination of wet and dry departures. The vertical blue line indicates the time of emergence, i.e. a permanent wet or dry departure. Example data is from the monthly total soil moisture content in the region, Northwestern North America (NWN), from the model ACCESS-CM2.

## 2.4 Green water planetary boundary: percentage of land area with dry and wet departures

The percentage of global ice-free land area in which root zone soil moisture departs from the Holocene baseline has been defined as the control variable for the green water planetary boundary (Wang-Erlandsson et al., 2022). For the reasons indicated in Section 2.1, we used total soil moisture and the pre-industrial baseline instead. We took the following steps in the calculation: first, the total land area of the ice-free climate regions, in which the monthly soil moisture content of the SSP scenario deviates from the PiControl baseline, was summed up for every model (including deviations that are not permanent). Second, this sum was expressed as a percentage of the total ice-free land area. Third, the median percentage of the ensemble was computed, so here we ended up with a monthly time series of the model ensemble median percentage of land area that departed from the baseline either as a dry departure or wet departure. Fourth, the 12-month rolling mean was calculated to more clearly show the trend of the projection from 2021 to 2100. Additionally, we calculated also the land area with soil moisture content departure that is permanent only, meaning only the land area with departures occurring after the time of emergence is included in the sum. Lastly, we repeated the land area aggregation for permanent wet and dry departures based on the yearly total soil moisture content.

## 3 Results and Discussion

### 3.1 Future deviations from the pre-industrial baseline

The mean total soil moisture content calculated from 80 years of PiControl data for 4 of the 14 different ESMs is shown in Fig. 2. The range of the PiControl total soil moisture content of each ensemble member appears to vary greatly from one another up to a difference of a factor of 10. However, the global distribution of soil moisture content seems to be quite similar. Also, it appeared that the deviations of the soil moisture content from PiControl to the different SSPs show quite similar magnitudes as illustrated by the regional deviation of each model (Fig. S3-6 in the supplement), which provides the argument for comparing ensemble median (Fig. 3).

In Fig. 3, it can be observed that, when going from SSP1-2.6 to SSP5-8.5, the deviation of the total soil moisture content from the baseline increases in most regions. In warmer scenarios there is also more drying than wetting (Fig. 4). Most of the regions in North and South America (excluding Northwestern North America (NWN) and Southeastern South America (SES)), southern Africa (West Southern Africa (WSAF) and East Southern Africa (ESAF)), and the Mediterranean become drier, while the regions in northern Africa (Sahara (SAH), Western Africa (WAF), Central Africa (CAF), Northeastern Africa (NEAF)), Eastern Europe (EEU), and Asia (excluding Tibetan Plateau (TIB) and East Asia (EAS)) become wetter as warming intensifies. The largest change occurs in the Northern South-America (NSA) region with a decrease in total soil moisture.

Warmer scenarios tend to amplify the signal that is already seen in the moderate scenarios, with some exceptions (West Central Asia (WCA), East Siberia (ESB), and South-East Asia (SEA)) that have contrasting wetting and drying signals for different scenarios. The general trends in Asia from SSP1-2.6 to SSP5-8.5 are not as consistent as in the other continents. The changes in direct neighboring regions occur in opposite directions. This trend continues and is more obvious in Section 3.2.

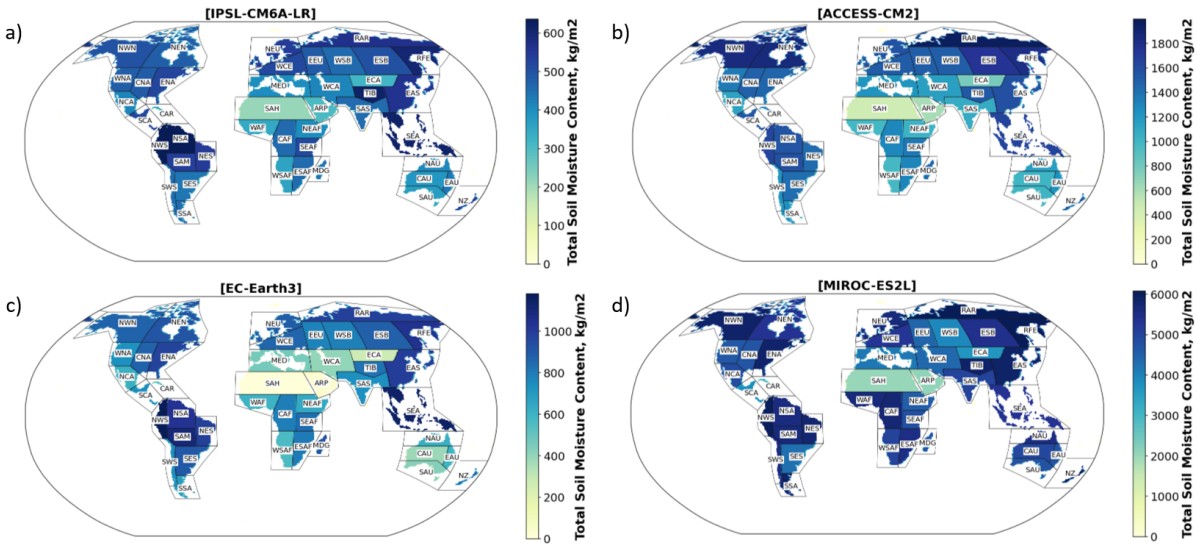

**Figure 2.** Examples of 80-year average values of the regional monthly total soil moisture content from the PiControl scenario in models IPSL-CM6A-LR (a), ACCESS-CM2 (b), EC-Earth3 (c), and MIROC-ES2L (d). (a) and (d) show the lowest and highest soil moisture contents among the ESMs respectively. The maps of other models are in included in Fig. S2 of the Supplement.

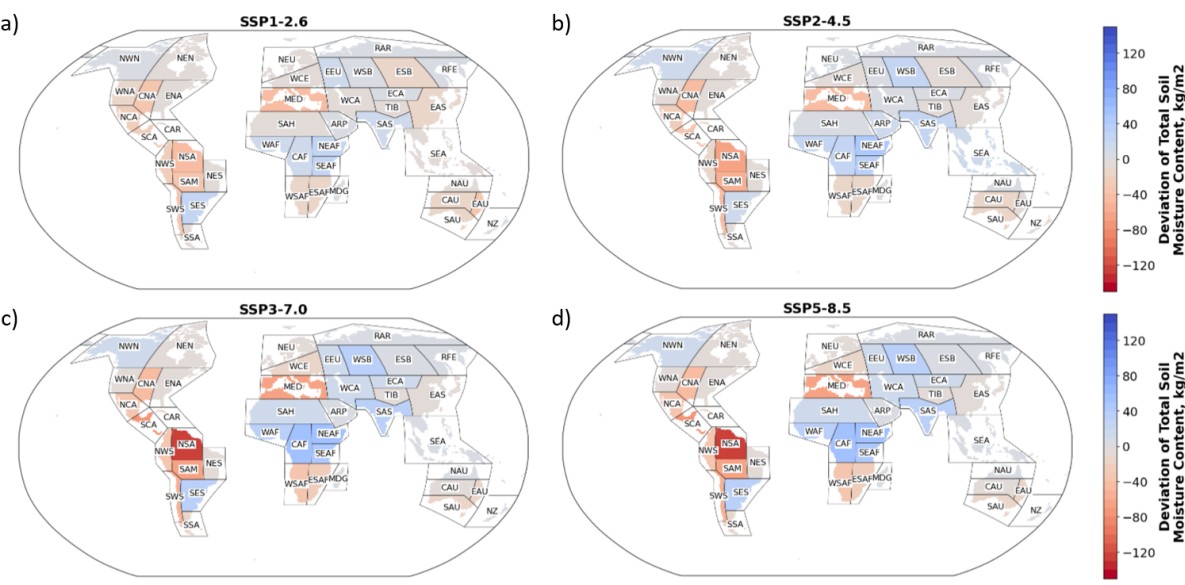

**Figure 3.** Deviation of the yearly mean total soil moisture content between 2071 and 2100 in each SSP from the PiControl scenario. The values depicted in the maps are the ensemble medians. The regions with a lower soil moisture content than the PiControl baseline are in red while the regions with higher soil moisture are in blue.

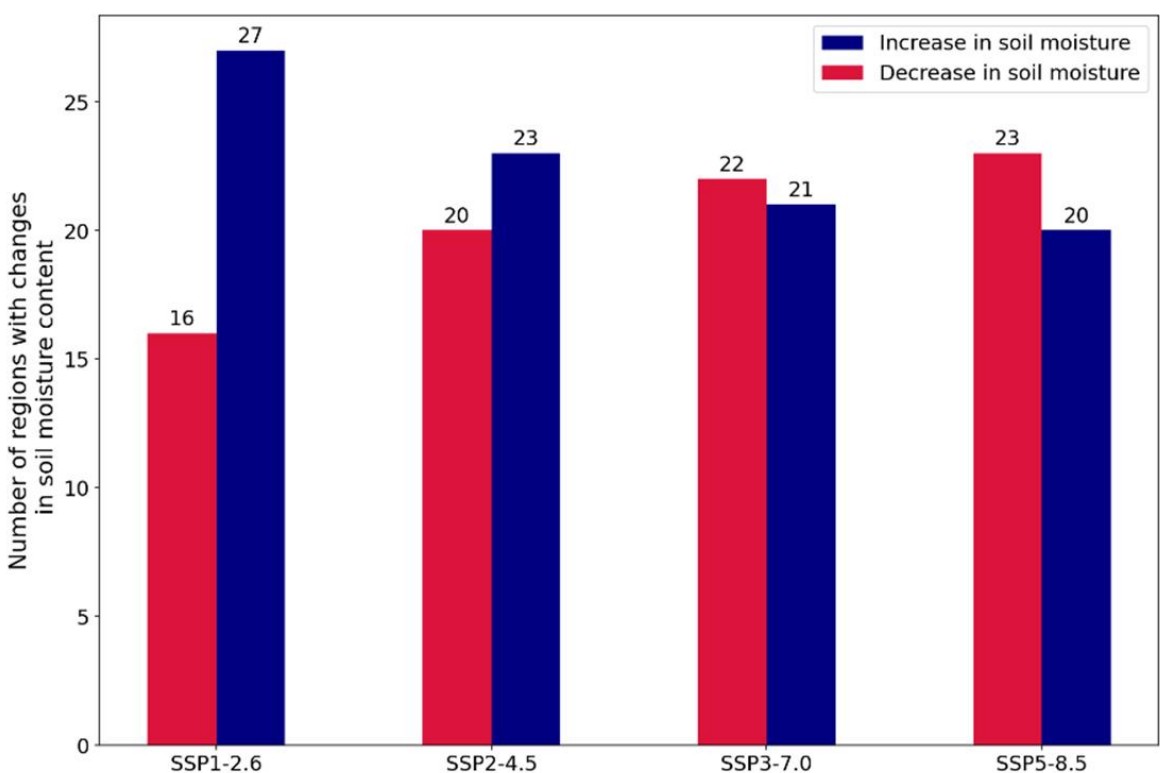

**Figure 4.** Number of regions with differences in yearly mean soil moisture contents between 2071-2100 and the PiControl baseline in different climate scenarios.

## 3.2 Time of emergence from the pre-industrial variability envelope

Looking at the percentage of models that show a permanent departure from the PiControl baseline (Fig. 5), we can observe a similar trend as for the deviation of total soil moisture content in Fig. 3. The percentage of models showing a permanent
departure from the PiControl baseline is higher in warmer scenarios. The Mediterranean (MED) has the highest percentage of models with a dry departure in SSP1-2.6 (approximately 70 %), and the percentage of models continues to increase to almost 100 % as it approaches SSP5-8.5. The North and South America (except NWN and SES) also see an increase in the percentage of models with a dry departure at different rates. The largest increases in dry departures occur in Northern South America (NSA) and Southwestern South America (SWS). On the contrary, there are obvious increases in the percentage of models with
a wet departure in some regions in Africa (SAH, WAF, NEAF, and CAF) and Asia (South Asia (SAS), West Siberia (WSB), and East Central Asia (ECA)) as warming intensifies.

Again, the inconsistency in the departure pattern in direct neighboring regions is seen in Asia. For example, as ECA experiences increasingly wet departure going from SSP1-2.6 to SSP5-8.5, East Asia (EAS) remains a region with dry departure, while TIB changes from having wet departure to dry departure. Some regions, such as South-East Asia (SEA) and Russian-

Far-East (RFE), experience a change from dry departure to wet departure as the effects of global warming strengthen although the changes are minor.

Fig. 6 illustrates the spread of the time of emergence for the regions in which more than half of the ESMs projects a permanent departure. In warmer scenarios, there are more regions with permanent departures. The spread of the time of emergence for regions with dry departure decreases from SSP1-2.6 to SSP5-8.5. The strongest signal for permanent dry departure is expected in MED, with a median expected time of emergence ranging from 2045 to 2075 for the different SSPs. However, according to some models the permanent departure in MED may have already happened and will continue even with the most rigorous mitigation measures. In the more pessimistic scenarios SSP3-7.0 and SSP5-8.5 several ESMs project some regions in South and Central America to have permanent dry departures around the middle of this century. However, permanent wet departures also occur for some African regions and the Middle East in the same time period. NEAF is expected to have the strongest signal for permanent wet departure, with a median expected time of emergence ranging from 2039 (SSP5-8.5) to 2095 (SSP1-2.6).

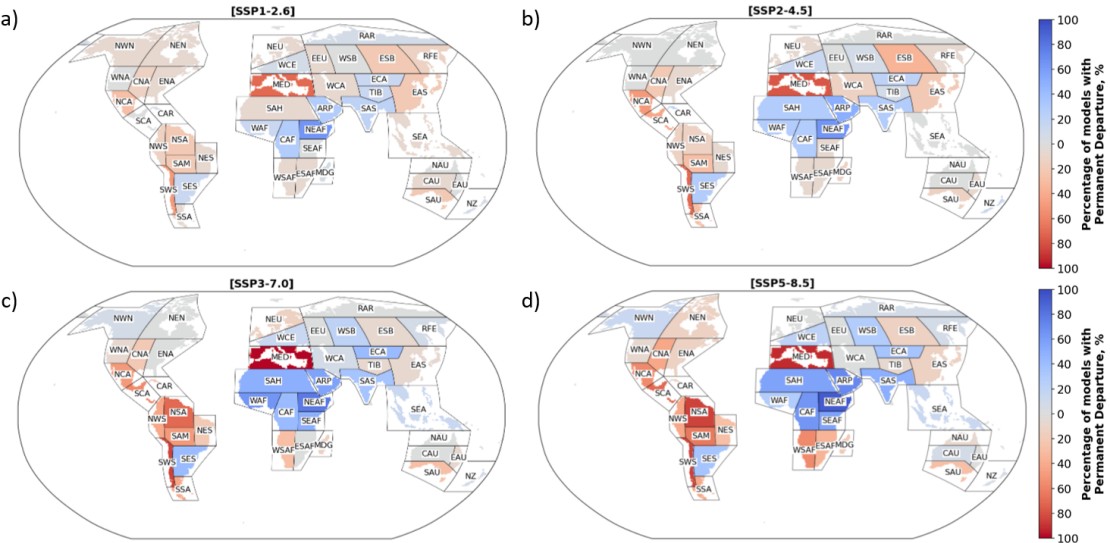

**Figure 5.** Percentage of models in each SSP scenario that show a permanent departure from the PiControl baseline between 2021 and 2100. The regions in blue are projected to experience a wet departure while those in red a dry departure. The exact regional time of emergence in each model is shown in Tables S2-5 in the supplement.

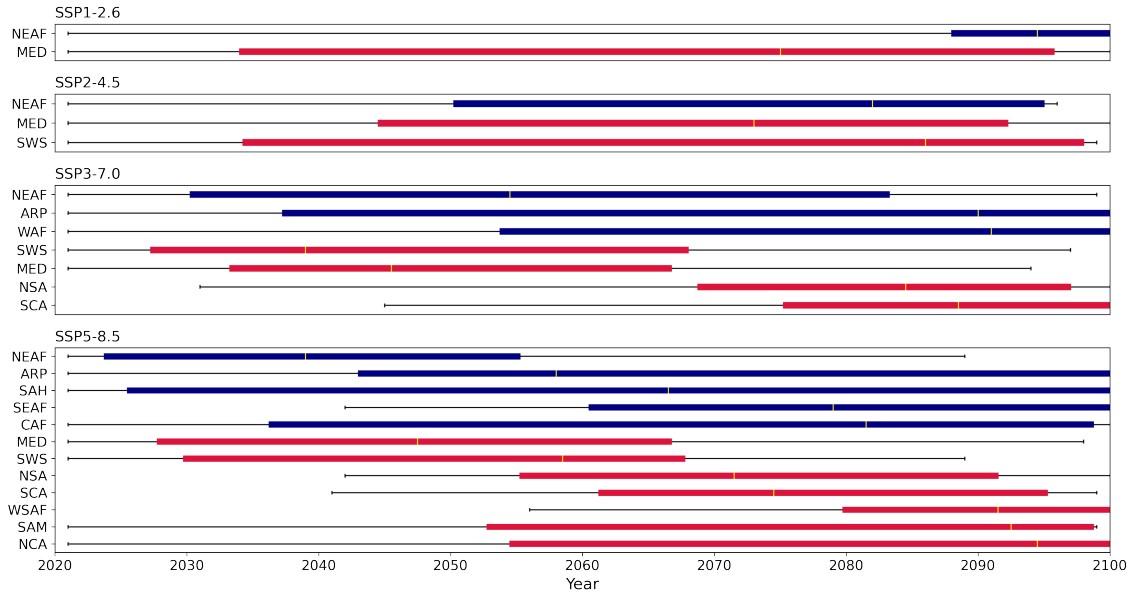

**Figure 6.** Distribution of the time of emergence of all ESMs for climate regions where more than 50 % of the ESMs show a permanent wet (blue) or dry (red) departure in different SSP scenarios within the studied timeframe. The interquartile range is represented by the length of the box, and median by the yellow line within the box.

## 3.3 Monthly departures from the pre-industrial climate

Fig. 7 shows that the regions in northern Asia and the northern North America have comparable numbers of months of wet and dry departure. Going from SSP1-2.6 to SSP5-8.5, the number of months in those regions increases slightly, meaning that extreme hydrological conditions are expected to occur more often. The most substantial drying is again detected in MED. In SSP1-2.6, 7 months (ensemble mean) in MED project a permanent dry departure by the end of the 21st century. This increases to 11 months (ensemble mean) in SSP5-8.5. The southern part of North America and South America, except SES, experience more serious drying with worsening warming.

Three southernmost regions in Africa, i.e. WSAF, ESAF, and Madagascar (MDG), experience an increase in the number of months with dry departures going from SSP1-2.6 to SSP5-8.5. However, the number of months with a wet departure increases in the northern part of Africa with the most increase occurring in NEAF.

Figures 8 and 9 show the regional number of months with a permanent departure in selected climate reference regions for each ESM in SSP1-2.6 and SSP3-7.0 respectively. The data for other climate regions and for SSP2-4.5 and SSP5-8.5 are included in Figs. S7-10 in the Supplement. As such, these figures highlight the differences between the results of each ESM. Although the results in many regions are diverse where some models show 12 months of wet departure and others 12 months of dry departure, a few regions have a higher agreement between different models, such as the regions from North Central-America (NCA) to WSAF in Figs. 8a and  9a. This finding is consistent with the regions that expect permanent departures in more than half of the ESMs studied (Fig. 6). The regions in Asia (e.g. from WSB to TIB in Figure 8a and 9a) and the

northern part of North America show more contrasting results between ensemble members. In warmer climate scenarios, the frequency of models with monthly permanent departures increases. The bar charts (Fig. 8b and 9b) indicate that among the climate regions that experience a permanent departure, many models project either a wet or a dry permanent departure for all 12 months in those regions.

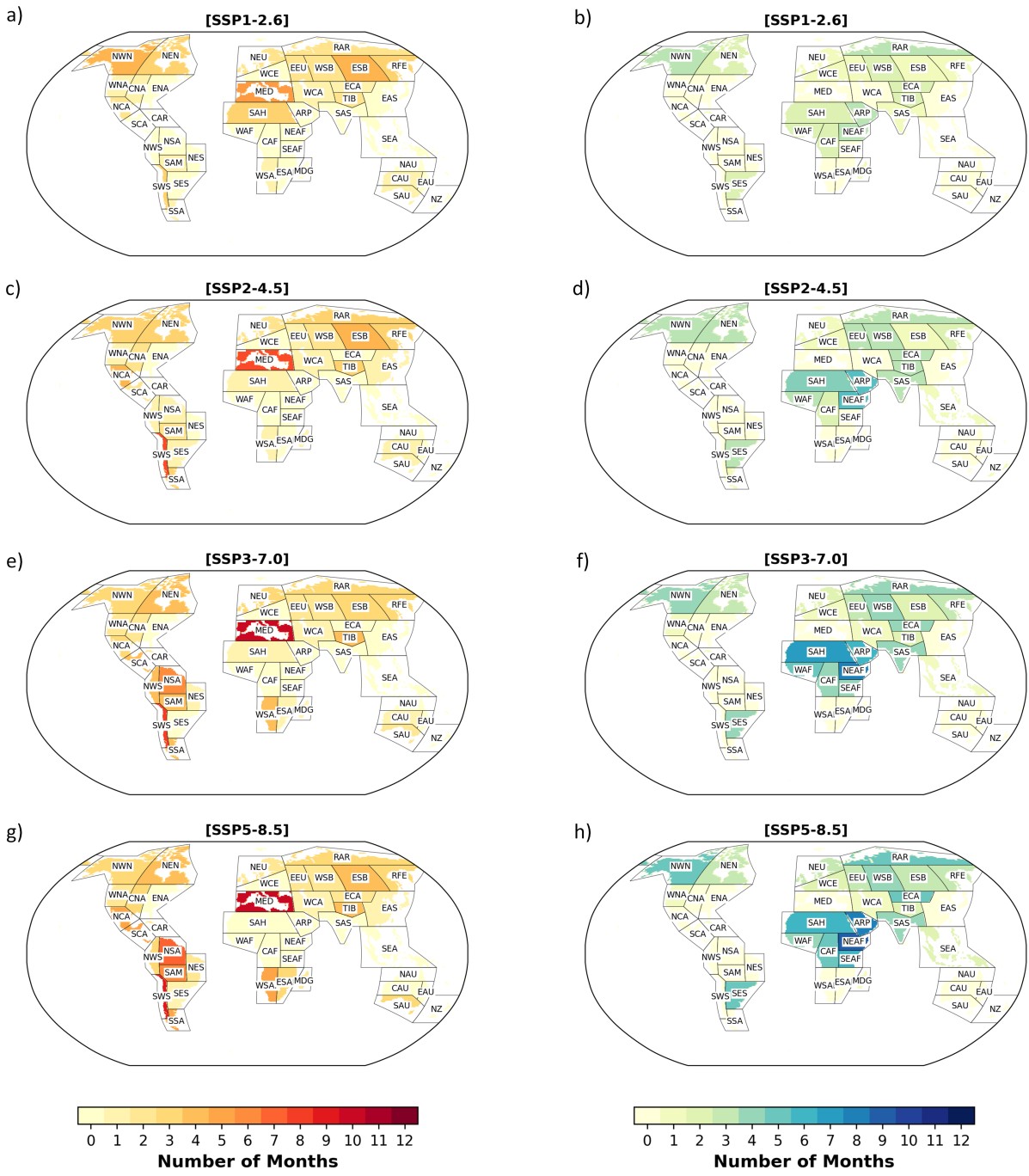

**Figure 7.** The ensemble mean number of months in which the monthly regional total soil moisture content deviates permanently from the PiControl baseline for each SSP scenario. Maps in a,c,e,g) show permanent dry departure for the respective SSPs, and b,d,f,h) permanent wet departures. The values shown in the maps are the regional ensemble means calculated from the numbers of months in each model (shown in Tables S6-13 in the Supplement). The number of months with a dry departure is shown in red, while the number of months with a wet departure is shown in blue.

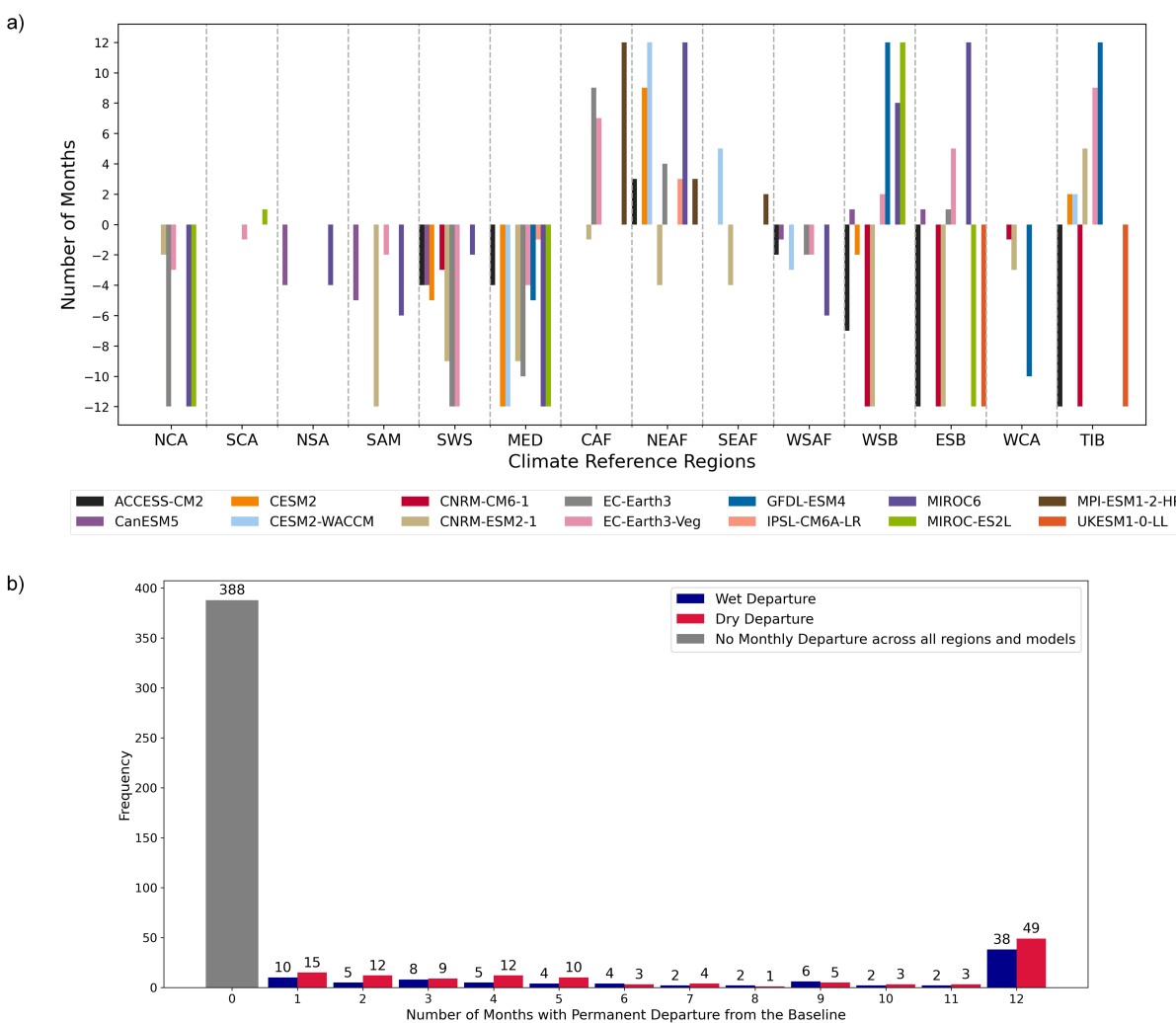

**Figure 8. (a)** For the SSP1-2.6 scenario: The number of months in which there is a permanent departure from the PiControl baseline variability for selected climate reference regions (x-axis). The data for other climate regions is available in Fig. S7 in the Supplement. The colors indicate different climate models. Negative values show dry departures, and positive values show wet departures. **(b)** Same data as in (a), but illustrating the frequency of the number of months with a permanent departure across each model and region. Zero indicates that a particular region in a model has no permanent departure at all.

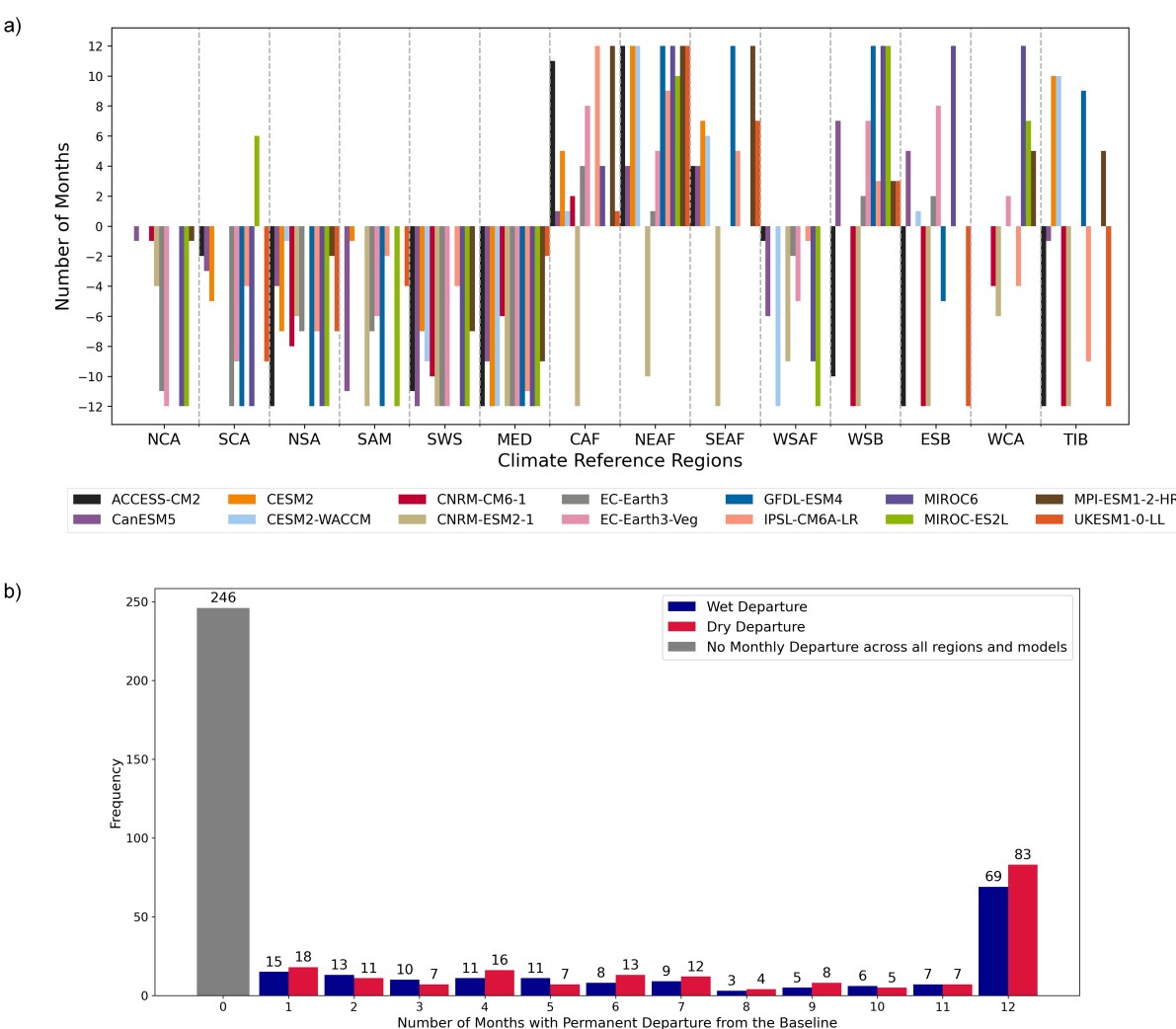

**Figure 9.** As in 8, but for the SSP3-7.0 scenario. The data for other climate regions is available in Fig. S9 in the Supplement.

## 3.4 Development of globally aggregated soil moisture change

Global land area with a wet or a dry departure increases from 2021 to 2100. Monthly departures in each SSP scenario (Fig. S11-18 of the Supplement), represented by the 12-month rolling mean in Fig. 10a,b, show the sum of all areas where the total soil moisture content is more than the 95th percentile or less than the 5th percentile regardless of whether the departure is permanent. The yearly permanent departures (Fig. 10c,d) show land area with a departure only after the regional time of emergence. For comparison, the 12-month rolling mean of the monthly departures and the yearly permanent departures for all four warming scenarios are plotted together in Fig. 10, and the comparison between SSP1-2.6 and SSP3-7.0 is shown in Fig. 11. In 2021, the land surface area with wet or dry departures shown in Fig. 10a,b is already higher than the expected percentage (5 % for both

wet and dry departures) in the case when climate was similar to pre-industrial conditions. This is consistent with Porkka et al. (2023) but higher in magnitude as there is a 15-year temporal gap and some methodological differences between the studies.

Fig. 10a shows that, in higher warming scenarios, the rate of increase in the land area with a monthly wet departure is higher. However, for the monthly dry departure (Fig. 10b), the differences between each SSP's rate of increase in the land area are smaller and less consistent with the warming trend. By the end of the 21st century, 22 % of the ice-free land in SSP1-2.6

experiences a wet departure and 28 % a dry departure, while 36 % of the ice-free land experiences a wet departure and 30 % a dry departure in SSP3-7.0. The trend of permanent wet departures (Fig. 10c) is again consistent with the warming trend, but this is not the case for permanent dry departures (Fig. 10d). In 2050, the percentages of the land surface area with permanent wet and dry departures (ensemble median) in SSP1-2.6 is expected to be 5 % and 4 % respectively, while in SSP3-7.0, these percentages are expected to be 14 % and 11 %. It can also be inferred that at least 15 % of the ice-free land is expected to

have permanently more soil moisture than the pre-industrial condition by the end of the 21st century as this is the result of the SSP1-2.6 scenario with the least warming effect, but could amount to 28 % in SSP3-7.0 (Fig. 11b). However, the results for dry departures show an opposite trend with 18 % of the ice-free land having permanently less soil moisture in SSP1-2.6 and 17.6 % in SSP3-7.0 due to a sudden rapid increase after 2090 in SSP1-2.6.

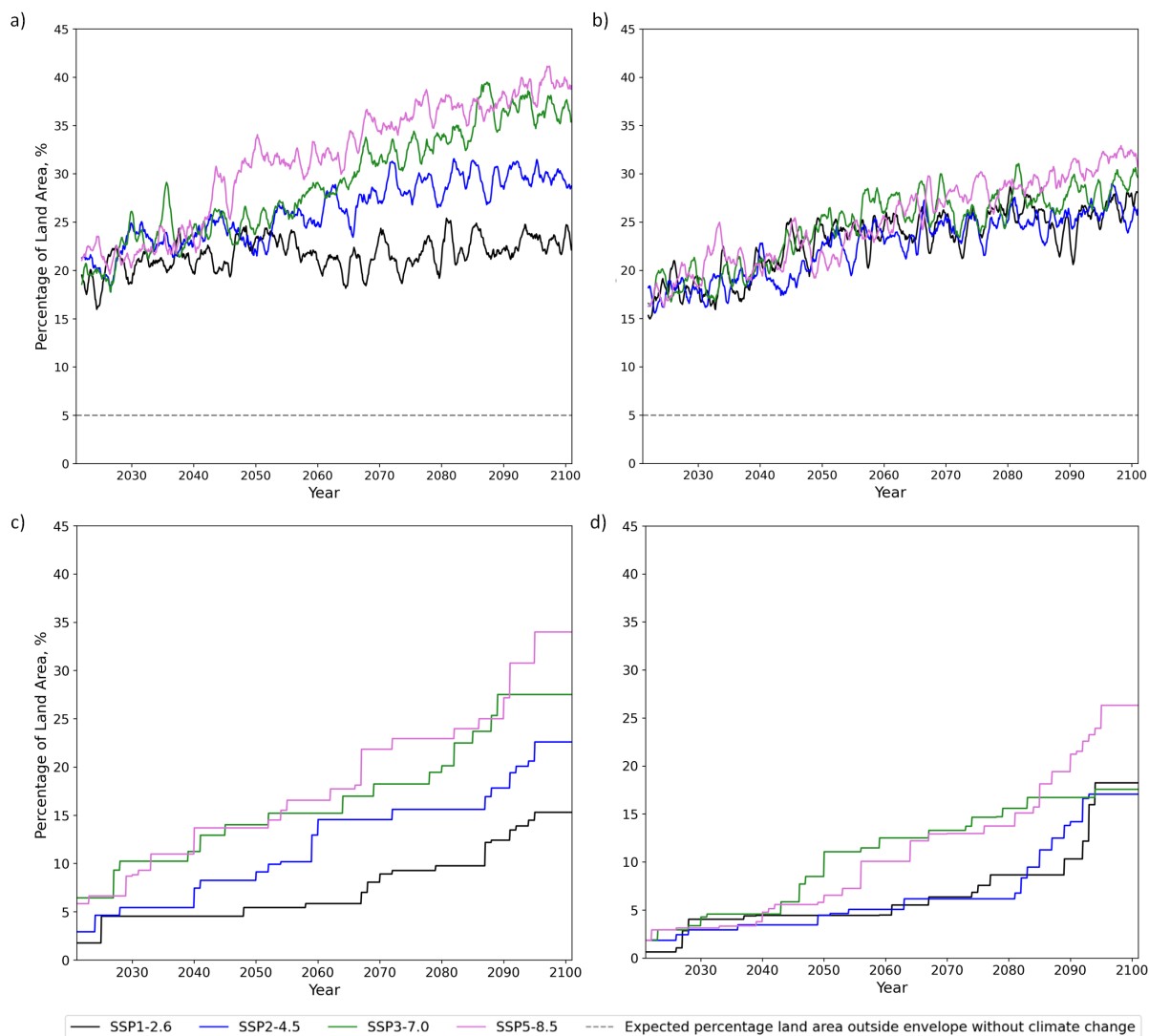

**Figure 10. (Top)** The monthly percentage of land area with soil moisture that exceeds the 95th **(a)** or goes below the 5th **(b)** percentile of the PiControl baseline in different SSP scenarios. **(Bottom)** The yearly percentage land area where there is permanent wet **(c)** or dry **(d)** departure from the PiControl baseline envelope in different SSP scenarios.

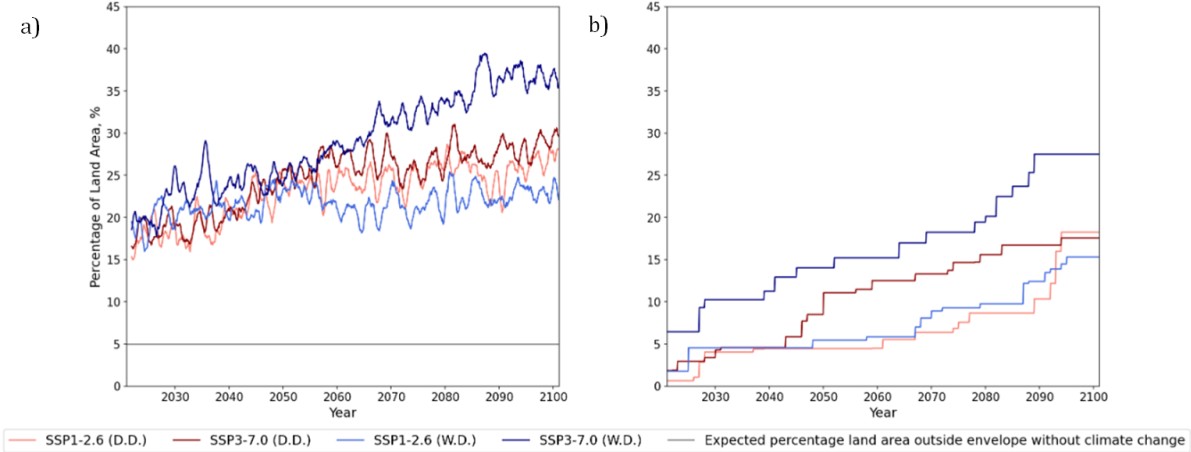

**Figure 11. (a)** The percentage of land area with a wet (in blues) or a dry (in reds) departure in SSP1-2.6 and SSP3-7.0 scenarios. **(b)** The percentage of land area where the wet (in blues) or dry (in reds) departure from the PiControl baseline is permanent in SSP1-2.6 and SSP3-7.0 scenarios. (D.D.= Dry Departure; W.D.= Wet Departure)

In general, the land surface area with a wet departure is projected to be larger than that with a dry departure for all SSPs, except SSP1-2.6, throughout the whole studied time frame. On the contrary, Dirmeyer et al. (2016) showed that permanently drier soil conditions are more globally prevalent than permanently wetter soil conditions. However, their study was based on the surface soil moisture (top 10 cm) content in CMIP5 with the historical simulation between 1860-2005 as the baseline. The surface soil moisture is more sensitive to increase in evaporation compared to the soil moisture content in the deeper layers which is more dominantly influenced by vegetation (Berg et al., 2017). This is reflected in Cook et al. (2020) that compared the variability of both surface and total soil moisture content between 2071-2100 and 1850-1880 (from historical simulation in CMIP6). Their study showed more widespread drying of surface soil moisture than total column soil moisture.

Our analysis on permanent departures indicate that a larger land area is predicted to have permanent wet departures. However, when investigating the differences between the yearly mean soil moisture content from 2071 to 2100 in different SSP scenarios and that of the PiControl scenario, we found that more regions will become drier rather than wetter in higher warming scenarios (Fig. 4). Yet, unlike permanent wet departures, this drying trend may not be permanent or may become permanent only at a later time, according to our definition of time of emergence. Therefore, the land area analysis (Figs. 10 and 11) detects a smaller land area with permanent dry departures than permanent wet departures.

Figure 12 compares the seasonal mean land area with permanent wet and dry departures in boreal summer (June-July-August, JJA) and boreal winter (December-January-February, DJF) for SSP1-2.6 and SSP3-7.0. For the latter scenario, the land area with permanent wet and dry departures in boreal summer is similar, but in winter, the land area with wet departures is much larger than for dry departures. In both SSP1-2.6 and SSP3-7.0, seasonal permanent wet and dry departures are comparable, but seasonal permanent dry departure remains higher in boreal summer than in boreal winter throughout the studied period. The seasonal soil moisture change in 2100 in Fig. 13 and the regional percentage of ESMs with permanent departures in different

seasons in Fig. 14 illustrate that there is drying of soil moisture in some regions (MED, CNA, SSA, SWS, SAM, WSAF, and

255 SAU) or wetting (SES, NEAF, SAS, WSB, and SAH) regardless of season. However, some other regions react differently depending on the season. High latitude regions in particular have a tendency to become more dry in summer and more wet in winter. This is in correspondence with Dirmeyer et al. (2016), but the signal is not very strong.

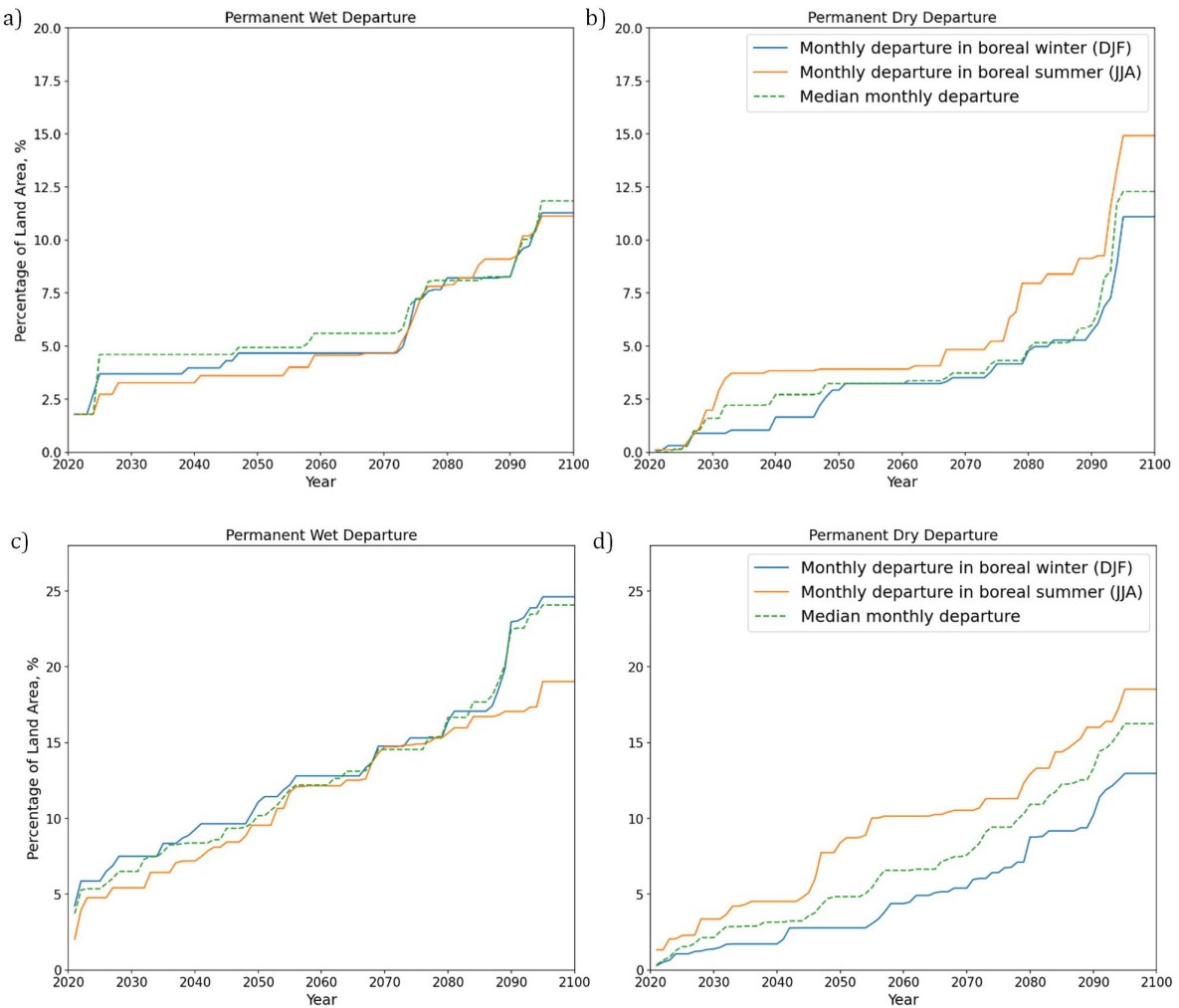

**Figure 12.** Seasonal mean land area with permanent soil moisture departures in months JJA (boreal summer) and DJF (boreal winter) in SSP1-2.6 ((**a**) and (**b**)) and SSP3-7.0 ((**c**) and (**d**)). The median monthly departure is the median of all 12 months (shown as the red dotted lines in Figs. S19 and S21).

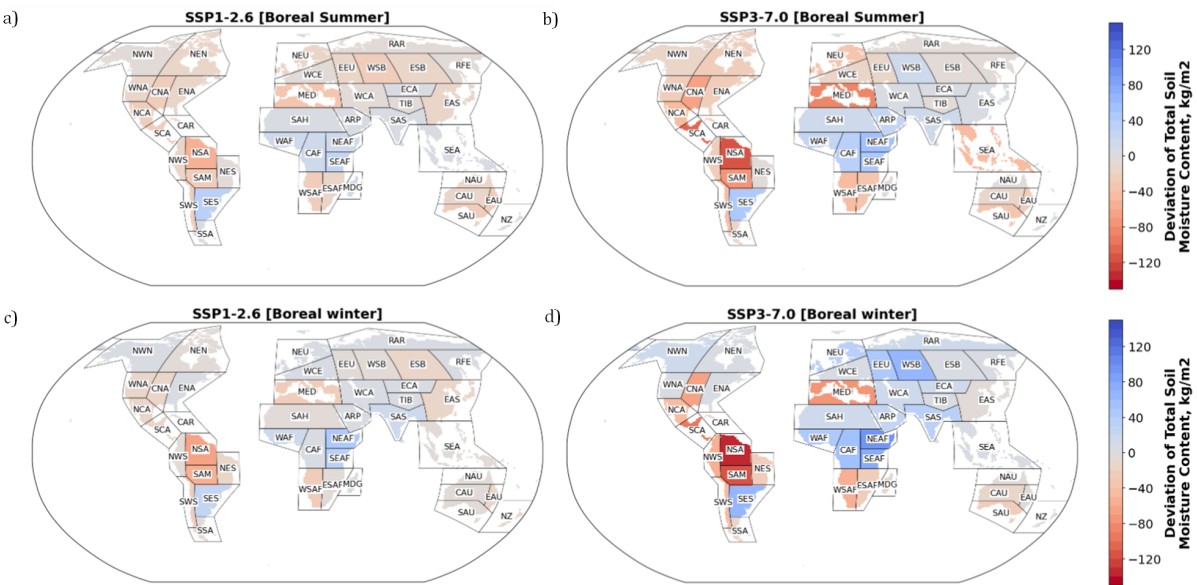

**Figure 13.** The difference of the seasonal mean total soil moisture content (values shown in the maps are the ensemble median) of SSP1-2.6 ((**a**) and (**b**)) and SSP3-7.0 ((**c**) and (**d**)) from the PiControl scenario in 2100. SSP2-4.5 and SSP5-8.5 are shown in Fig. S23 of the Supplement.

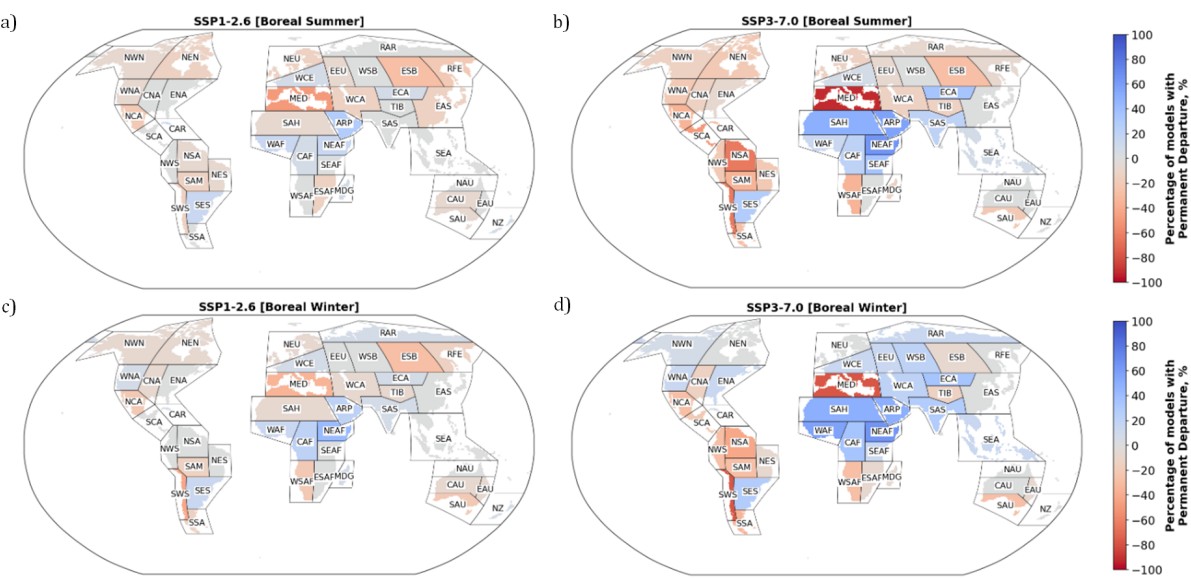

**Figure 14.** Percentage of models in SSP1-2.6 ((**a**) and (**b**)) and SSP3-7.0 ((**c**) and (**d**)) that show a permanent departure from the PiControl baseline in different seasons in 2100. SSP2-4.5 and SSP5-8.5 are shown in Fig. S24 of the Supplement.

Drying and wetting of soils may be the result of different combinations of changes in the water balance terms precipitation, runoff and evaporation. A quantitative analysis of each of these terms was considered out of scope. Dirmeyer et al. (2016) showed that the top 10 cm of soil moisture changes correlate well with precipitation changes, although drying of soil moisture also happens despite increased precipitation in mid- and high-latitudes in summer. Drying of surface soil moisture, however, was also found to be consistent with enhanced evaporative demand (Berg et al., 2017). Cook et al. (2020) showed regions (e.g., extra-tropical South America, northern and eastern Africa, India, and Central Asia) with robust increases in annual soil moisture (both surface and total column) that are consistent with areas that are projected to experience the strongest increases in precipitation. However, they reported that drying in the total soil moisture column is spatially more widespread compared to precipitation and runoff, and drying can appear in regions where precipitation is increasing (e.g. northern and eastern Europe), which is the result of greater evaporative demand (Dai et al., 2018; Mankin et al., 2019). Regarding the soil moisture changes in our study, it makes sense to assume that precipitation changes are the dominant driving mechanism, but changes in evaporative demand can reduce the effect of precipitation changes in some regions.

## 3.5 Limitations

Here, we would like to stress that our results should be interpreted by taking into account the methodological choices and assumptions made in this study. The most important limitations are:

- The uncertainty associated with the capacity of the ESMs to project the water cycle. We already observed a large model spread (Fig. 8 and 9), and we did not weigh our models based on performance or interdependence. However, this usually does not dramatically influence the results (e.g., Brunner et al., 2019). Moreover, model spread should not only be considered as errors made in the modeling assumptions, but as such, it also reflects the wide range of potential futures given our limited understanding of the Earth system Jebeile and Barberousse (2021). The IPCC regards model spread as something that merely reflects the quantification of previously unmeasured sources of uncertainty (Stocker, T. et al., 2013, IPCCFAQ 1.1) and according to Mankin et al. (2019), convergence of ensembles of ESM projections has lower priority than model independence that is a prerequisite for the robustness of models' core hypotheses. Model spread by model independence may not come from detrimental dissensus among ESMs (which would undermine confidence), but from different yet equally valid hypotheses and mechanics in ESMs. Moreover, ensemble means tend to be more accurate than projections from individual models, because of independence among ESMs that stems from divergence among ESMs (Reichler and Kim, 2008). Therefore, our model spread is not detrimental to the presented analyses.

- ESMs have particularly high uncertainties in certain regions, such as the Sahel (Monerie et al., 2020) where we found mostly wet departures. Thus, given the ESM limitations in such regions and the lack of observational in-situ data for validation, more research is needed to firmly establish such projections regionally. Another example is the Amazon, where misrepresenting land-atmosphere interactions could lead to distinctively different rainfall patterns in future projections (Baker et al., 2021). In total, the percentage of regions with conflicting departure signals globally is 33 % for SSP-3.70, and 37 % for SSP-1.26. Most of the regions in which signals are conflicting signals are at higher northern latitudes.

Similarly large model spread has been observed since CMIP5 (Berg et al., 2017). Fig. S25 in the Supplement shows the percentage of regions around the latitudes, 60°N, 40°N, 20°N, 0°, 20°N, and 40°S, with conflicting departure signals among ESMs in each SSP scenario.

– The land cover and its change in most ESMs is prescribed (Hurtt et al., 2020), hence the root zone soil moisture may undergo dynamics due to land cover adaptation and land cover change (Singh et al., 2022) that may be better captured by more sophisticated vegetation models (e.g., Sakschewski et al., 2021), yet offline simulations with such models would also come with the limitation of water and energy balance inconsistencies with the forcing data.

– ESMs respond differently to the increase in atmospheric carbon dioxide in future climate scenarios, which further propagates to their estimates of hydrological variables. The range of effective climate sensitivity (ECS) in CMIP6, 1.8 to 5.6 K (Zelinka et al., 2020), is larger than that of CMIP5, 2.1 to 4.7 K (Andrews et al., 2012), and the historical ECS range, 1.5 to 4.5 K (Council, 1979). A few models which are used in our study, including CanESM5, CESM2, UKESM1, and CNRM-CM6-1, notably have ECS values exceeding the upper limit of the ECS range in CMIP5 (Zelinka et al., 2020), which likely results from stronger positive cloud feedback and the combination of weak overall negative feedback and moderate radiative forcing (Zelinka et al., 2020). The high ECS values in some models could mean more widespread and pronounced departures of soil moisture predicted in these models compared to what would be realistic.

– Permanence is here defined as departure without reversal until the year 2100. Clearly, longer simulation periods beyond 2100 might affect the estimations of areas with permanence and the associated time of emergence. Future research might want to investigate the impact of emission trajectories on extent of permanent departures beyond 2100. Besides, the time of emergence is sensitive to small difference as little fluctuations that cause the soil moisture content to fall back within the baseline envelope will affect the time of emergence.

– While we aggregated our data using the IPCC climate reference regions, which are defined based on climate homogeneity distinguished by multiple factors (Iturbide et al., 2020), any spatial aggregation loses a certain degree of information. Hence, aggregating spatial means over regions could lead to over- or underestimation of the percentage of land area with departures because it is sensitive to extreme values. This can especially be the case in regions in which major human impacts (e.g. extensive irrigation or drainage) cover a relatively small geographical area but change the regionally aggregated mean soil moisture substantially.

– In some regions, the soil moisture trends concur with expected impacts e.g. agriculture, such as in southern Europe, where severe impacts on crop production is projected (Cramer et al., 2018). In others, a wetting trend might not be directly coupled to crop productivity, since it also critically depends on e.g. temporal variability and distribution of precipitation and temperature (e.g., Porkka et al., 2021), as well as availability of irrigation options (e.g., Elliott et al., 2014).

– The wet departures in very dry regions such as the Sahara, (Figs. 5 and 7), indicate that the soil moisture in those regions is expected to be permanently higher than in the PiControl baseline according to our analysis based on relative

change. However, the absolute changes in soil moisture are of rather low magnitudes and the implications regarding
water availability and the ecosystem response for such extremely arid regions could be rather limited.

  – The interpretation of the results can be different when looking at ensemble means or medians for the analysis in section
    3.4. In the analysis for land areas with permanent departures (Fig. 11 & 10), the results are reported as ensemble medians.
    Fig. S26 in the Supplement shows the difference between mean and median land area with permanent departures for
    SSP1-2.6 and SSP3-7.0. If the results are reported in mean, the difference between land area with wet and dry departures
    is smaller. However, when the results from each model are investigated individually, there could be outliers, such as in
    this case the model CNRM-ESM2-1 that reports much higher dry departures than the others.

## 4  Conclusions

In this study we analyzed the future (2021-2100) changes in total soil moisture in different IPCC reference regions with respect
to the pre-industrial baseline (1850-like conditions). Four different scenarios (SSP1-2.6, SSP2-4.5, SSP3-7.0, and SSP5-8.5)
from 14 ESMs in CMIP6 were studied. We evaluated different aspects: the deviation of total soil moisture from the PiControl
scenario, permanent departures beyond the PiControl variability and the time of emergence of those permanent departures, on
both yearly and monthly scales. Despite some ESM disagreement, we found several clear and consistent signals:

  – All indicators showed remarkable drying in the Mediterranean, South Africa, southern North America, and South Amer-
    ica (except the region of South-Eastern South America). By 2050, in the Mediterranean, 43 % of the ESMs show perma-
    nent dry departure in SSP1-2.6 and 57 % in SSP3-7.0. By 2100, the percentages increase to 70 % in SSP1-2.6 and 100 %
    in SSP3-7.0.

  – Considerable wet departures were detected in Northern Africa, South-Eastern South America, and Southern Asia.

  – The agreement between the ESMs in the above-mentioned regions with strong departure signals, wet or dry, is higher
    than the other regions as indicated in Section 3.3.

  – The magnitude of drying and wetting increases with the effect of global warming in most regions with some exceptions
    of a few regions in northern North America, Asia, and Australia.

  – The regional departures in Asia were not as large and consistent as in the other continents and the agreement between
    ESMs is also lower there.

  – By the end of the 21st century, the percentage of the land surface area with permanent wet departures from the pre-
    industrial soil moisture variability is expected to be: 15 % in SSP1-2.6, 23% in SSP2-4.5, 28 % in SSP3-7.0, and 34 %
    in SSP5-8.5

  – The percentage of the land surface area with permanent dry departures: 18 % in SSP1-2.6, 17 % in SSP2-4.5, 17.6 %
    in SSP3-7.0, and 26 % in SSP5-8.5.

- The land area with wet departures (both permanent and non-permanent) and its rate of increase are slightly larger than that with dry departures for all SSP projections, except SSP1-2.6, throughout the studied timeframe.

These analyses show that anthropogenic warming and land system change captured in the SSP scenarios constitute crucial drivers of transgression of the planetary boundary of green water. The results show that the planetary boundary of green water might be more transgressed than previously thought (Wang-Erlandsson et al., 2022), and that the transgression will likely increase regardless of emissions trajectory. Even in the SSP1-2.6 trajectory in which the most vigorous climate mitigation efforts take place, the total soil moisture content in 33 % of the ice-free land surface area is anticipated to permanently depart from the pre-industrial variability by the end of the 21st century. In about 10 % of the land areas, permanent departures can be expected already by 2050. Hence, even with global mitigation efforts, these results call for swift adaptation to the new hydrological situation in several regions around the globe. However, the transgressions will be considerable more serious and widespread in higher emissions scenarios (e.g. in SSP3-7.0, 25 % land area with permanent departure), pointing to the critical role of mitigation.

We further note that emissions trajectory can make a considerable difference for the degree of permanent drying and wetting in major regional weather systems or tipping elements of the Earth system (Lenton et al., 2019, fig. 4). For example, high emissions scenarios are expected to lead to substantial drying in the Amazon forest and substantial wetting in the land areas of the Indian and West African monsoons. The concepts of permanence and time of emergence could potentially be used in future research to better understand temporal dimensions of Earth resilience and the recoverability of planetary boundary transgressions.

*Code availability.* The code for data processing and visualization is available at https://doi.org/10.5281/zenodo.8166198

*Data availability.* Data underlying the figures and tables presented in this paper can be generated by running the code available from https://doi.org/10.5281/zenodo.8166198

*Author contributions.* Conceptualization: RvdE. Methodology: EL, RvdE. Software: EL. Formal Analysis: EL, RvdE. Data Curation: EL. Writing – Original Draft: EL, RvdE. Writing – Review & Editing: EL, LW-E, VV, MP, RvdE. Visualization: EL. Supervision: RvdE.

*Competing interests.* The authors declare that they have no competing interests.

*Acknowledgements.* CMIP6 ESM data used in this study were made freely available from the Earth System Grid Federation (ESGF) website (https://esgf-node.llnl.gov/projects/cmip6/; ESGF (2022)). Specifically, we acknowledge the following contributions to the CMIP6 archive:

Pre-industrial control simulations (Hajima et al., 2019; Swart et al., 2019b; Boucher et al., 2018; EC-Earth Consortium, 2019b; Danabasoglu, 2019c; EC-Earth Consortium, 2019a; Danabasoglu et al., 2019; Dix et al., 2019a; Jungclaus et al., 2019; Krasting et al., 2018; Seferian, 2018; Tang et al., 2019; Tatebe and Watanabe, 2018; Voldoire, 2018) and Scenario Model Intercomparison Project (ScenarioMIP) (Dix et al., 2019b; Swart et al., 2019a; Shiogama et al., 2019; Tachiiri et al., 2019; Danabasoglu, 2019b, a; Consortium , EC-Earth; Boucher et al., 2019; Consortium , EC-Earth; Schupfner et al., 2019; Voldoire, 2019a, b; Good et al., 2019; John et al., 2018). The CMIP6 ESM data was downloaded from

Pangeo's Google Cloud into a Python programming environment for the analyses (http://gallery.pangeo.io/repos/pangeo-gallery/cmip6/). The updated IPCC reference regions were available from (https://github.com/SantanderMetGroup/ATLAS; Iturbide et al. (2020)). RE acknowledges funding from the Netherlands Organisation for Scientific Research (NWO), project number 016.Veni.181.015. LWE acknowledges financial support from the European Research Council through the 'Earth Resilience in the Anthropocene' project (no. ERC-2016- ADG 743080). We would like to thank the editor, Harrie-Jan Hendricks Franssen, referee, Wolgang Wagner, and two anonymous referees for

providing constructive comments that helped to improve our manuscript.

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
