# Peer review of "Root zone soil moisture in over 25 % of global land permanently beyond pre-industrial variability as early as 2050 without climate policy"

_EGUsphere, 2022_

## Author Response (AR1)

**Responses to all referees**

This document contains the final responses to all three referees. Most answers are the same as in the online discussion, but some have been updated to reflect the final updates in the revised manuscript

We cite the referee comments in black. We added numbers for clarity purposes.

Our responses are in blue

**Content**

**Response to Referee 1**

"This study analyzed the future changes of the total soil moisture for the four different SSP scenarios. Using 14 ESM in CMIP6, deviation of total soil moisture from the PiControl scenario, permanent departures beyond the Picontrol variability and the time of emergence of those permanent departures are evaluated. Also, this study analyzed regional total soil moisture variability results according to the SSP scenarios in detail and presents the robust results of multi ensemble ESM in some regions like Mediterranean in terms of remarkable dry departure. Regions such as Northern Africa, South-Eastern South America and Southern Asia resulted in considerable wet departures. In many regions, these dry and wet intensities displayed to intensify as the effects of global warming. The priority of this study is to quantitatively organize the regional results according to the SSP future scenarios over global domain. However, as mentioned in chapter 3.5, there are many limitations in relation to the analysis of future SSP scenario results, and additional explanations on the results and data seem to be needed. Detailed comments are below."

We thank the referee for a thorough review and constructive comments. We hope to satisfy the referee by adding the requested additional explanations in a revised manuscript.

**Major Comments:**

1.  Regarding the title, this study expressed root zone soil moisture in over 25% of global land permanently beyond pre-industrial variability as early as 2050. However, in practice, the analysis of total soil moisture is the main focusing variable in ESM in CMIP6, and since the percentage value differs depending on the scenario, it is necessary to modify the tilt (title?) to reflect these aspects.

    Regarding the aspect of total vs. root zone soil moisture, we believe this is sufficiently explained in section 2.1 Data: "The total soil moisture content is the mass of water in all phases and in all soil layers. Whereas the green water planetary boundary definition is based on root zone soil moisture (Wang-Erlandsson et al., 2022), this was not explicitly available from most ESMs in CMIP6. Depending on the model configuration the total soil moisture may coincide with root zone soil moisture in most areas (e.g., van Oorschot et al., 2021). In any case, we deem it logical to assume that any changes occurring in the total soil moisture in fact are occurring in the hydrological active zone, which is the zone in which plant roots are active (e.g., Feddes et al., 2001; Fan et al., 2017; Singh et al., 2020), and, therefore, we focus on analyzing the absolute and not the relative changes in total soil moisture."

Regarding the aspect of different change depending on the scenario, we intended this to be read as something which could happen without climate policy, however, this was apparently not intuitive for a reader, hence we decided to follow the referee's suggestion and propose to change the title to:

"Root zone soil moisture in over 25% of global land permanently beyond pre-industrial variability as early as 2050 without climate policy".

The 25% land area with departure occurs in the SSP3-7.0 scenario in which there is no climate policy implemented. To the sentence "The SSPs used in this study are SSP1-2.6 (+2.6 W m$^{-2}$; low GHGs emissions), SSP2-4.5 (+4.5 W m$^{-2}$; intermediate GHGs emissions), SSP3-7.0 (+7.0 W m-2; high GHGs emissions), and SSP5-8.5 (+8.5 W m-2; very high GHGs emissions) (Riahi et al., 2017).", we added:

"Note that SSP2-4.5 is roughly on the current pathway of emission reductions and SSP3-7.0 is an average 'no climate policy'-scenario (Hausfather and Peters, 2020)"

2. As mentioned in the limitations part of this text, ESM have uncertainties in certain regions. For example, in this study, North Africa like SAH desert regions are historically very dry but tend to show wet departures in the future. it comes out as a wet case because the 95% percentile wet departure threshold is low in very dry regions like dessert, it seems more necessary to reflect the climatological soil moisture distribution and land type since the consequences that desert areas becomes wet are considered unacceptable.

We agree that a wet departure in a very dry region such as the Sahara may be a small increase in absolute terms due to the low threshold, but departure does happen indeed. Since we are analyzing comparisons of soil moisture with the baseline, the conclusion is in relative terms. We intended to report our results neutrally and we did not mean to imply that such a wet departure is by definition unacceptable. We added this point of low absolute changes in the limitations:

"The wet departures in very dry regions such as the Sahara, (Figs. 5 and 7), indicate that the soil moisture in those regions is expected to be permanently higher than in the PiControl baseline according to our analysis based on relative change. However, the absolute changes in soil moisture are of rather low magnitudes and the implications regarding water availability and the ecosystem response for such extremely arid regions could be rather limited."

3. Also, some regions show contrasting wetting and drying signals for different scenarios, which shows a high regional uncertainty according to ESM, which makes the results less reliable. It seems reasonable to add an analysis to the results by latitude or by representative land type.

We agree that convergence of ESMs projections intuitively increases confidence in model projections and that conflicting model outcomes reduce that confidence regionally. However, this is not the full story. Jebeile & Barberousse (2021) wrote that model spread is indeed an indication of models' uncertainty, but should not only be considered as the errors made in the modeling assumptions. The IPCC regards model spread as something that merely reflects the quantification of previously unmeasured sources of uncertainty (Stocker and Qin, 2013, IPCCFAQ 1.1). According to Jebeile & Barberousse (2021), regarding ESMs ensembles, convergence of projections has lower priority than model independence, which is a prerequisite for the robustness of models' core hypotheses. Independence may not come from detrimental dissensus among ESMs that undermine confidence, but from different yet equally valid hypotheses. Moreover, ensemble means tend to be more accurate than projections from individual models, because of independence among ESMs that stems from divergence among ESMs (Reichler & Kim, 2008). Therefore, model spread is not necessarily detrimental to climate impact assessments. This is now more extensively discussed in the manuscript. Not copied here, but please see the limitations section and note that for each region the supplementary material shows to what extent the ESMs have contrasting signals.

4. Regarding total soil moisture analysis using different ESMs in CMIP6, future scenario results have forcing and ESM dependency issues. Therefore, in order to derive general results, it seems necessary to understand and explain how the amount of total soil moisture changes in terms of precipitation and run off in terms of water balance. In this study, a detailed regional analysis of dry and wet conditions was presented in detail, but explanations for the reasons for the results are considered insufficient. A scientific understanding would be better if given an additional explanation of energy balance or water balance for the variation of soil moisture.

We understand that soil moisture data depends on different ESMs and their forcings. However, as mentioned in the previous part, the model spread in the ESMs also shows that there is model independence, which is not necessarily detrimental for a more accurate ensemble mean. Drying and wetting may be the result of different combinations of changes in the water balance terms precipitation, runoff

and evaporation. We agree that explanations for the reasons behind the changes would give the reader a more comprehensive picture about the impact of climate change on soil moisture. However, this may also be regionally dependent and we hope it is understood that a detailed analysis of all regions is not feasible in this paper. We added the text below to the revised manuscript at the end of Section 3.4:

"Drying and wetting of soils may be the result of different combinations of changes in the water balance terms precipitation,runoff and evaporation. A quantitative analysis of each of these terms was considered out of scope. Dirmeyer et al. (2016) showed that the top 10 cm of soil moisture changes correlate well with precipitation changes, although drying of soil moisture also happens despite increased precipitation in mid- and high-latitudes in summer. Drying of surface soil moisture, however, was also found to be consistent with enhanced evaporative demand (Berg et al., 2017). Cook et al. (2020) showed regions (e.g., extra-tropical South America, northern and eastern Africa, India, and Central Asia) with robust increases in annual soil moisture (both surface and total column) that are consistent with areas that are projected to experience the strongest increases in precipitation. However, they reported that drying in the total soil moisture column is spatially more widespread compared to precipitation and runoff, and drying can appear in regions where precipitation is increasing (e.g. northern and eastern Europe), which is the result of greater evaporative demand (Dai et al., 2018; Mankin et al., 2019). Regarding the soil moisture changes in our study, it makes sense to assume that precipitation changes are the dominant driving mechanism, but changes in evaporative demand can reduce the effect of precipitation changes in some regions."

5. This study result presents the land surface area with a wet departure is projected to be larger than that with a dry departure for SSP scenarios. As mentioned, this result was confirmed by Dirmeyer et al. (2016), it is different from the contents of drier soil condition are more globally prevalent, and it is explained that the influence of vegetation is large when using total soil moisture. According to Dirmeyer et al. (2016), a seasonal difference was also reported that JJA became drier in summer but wet in winter. In this study, it seems necessary to include a discussion that reflects the seasonal cycle, and further explanation is needed on whether the results are robust in terms of annual mean calculation and how the effect of vegetation on total soil moisture is reflected in the results in detail.

We are not 100% certain that we understand the point of the referee. We think the referee may refer to the fact that wet departures in winter and dry departures in

summer could cancel each other out. In fact we have also considered this possibility and hence we based much of our analysis on monthly data. So, for example, in Figures 6-8 (original manuscript) any region can have months with both permanent wet and dry departures and, if so, these results are visualized as such. The plots in Figs. C1 and C2 below show the percentage of regions around the latitudes, 60°N, 40°N, 20°N, 0°, 20°N, and 40°S, with conflicting departure signals among ESMs. The percentage of regions with conflicting departure signals globally is 33% for SSP370, and 37% for SSP126. Most conflicting signals are at higher northern latitudes. Similar phenomenon (i.e. larger spread) has been observed since CMIP5 (Berg et al., 2017). This discussion was added into the Limitations section of the revised manuscript, and the Figs. C1 and C2 were added to the Supplement.

[Figure]

Figure C1. Percentage of regions that show conflicting wet and dry departure signals among ESMs around different latitudes.

[Figure]

Figure C2. As in Fig. C1 but for SSP3-7.0

Figures C3 and C4 below compare the seasonal mean land area with permanent wet and dry departures in boreal summer (June-July-August, JJA) and boreal winter (December-January-February, DJF) for SSP1-2.6 and SSP3-7.0. For SSP3-7.0, the land area with permanent wet and dry departures in boreal summer is similar. However, the land area with wet departures is significantly larger than dry departures in boreal winter. This analysis will be added to the manuscript under the Result and Discussion section along with seasonal plots (analysis in progress) that illustrate the regions of wet and dry departures in different seasons at the end of the 21st century.

[Figure]

Figure C3. Land area with permanent departures for months DJF and JJA in SSP1-2.6. The median monthly departure is the median of all 12 months.

[Figure]

Figure C4. As in Fig. C3 but for SSP3-7.0.

The two figures (Figs. C5 and C6) below show the differences between the land area with permanent departure of soil moisture for each month in SSP3-7.0. Each solid line represents the ensemble median of a particular month. The red thick dotted line shows the median. These figures were added to the revised Supplement along with similar figures for other SSP scenarios.

[Figure]

Figure C5. Land area with permanent wet departure of soil moisture in SSP3-7.0 based on monthly analysis.

[Figure]

Figure C6. As in Fig. C5 but for dry departure.

Although our analysis on permanent departures indicate that a larger land area is predicted to have permanent wet departures, our first analysis, that investigates the difference between the yearly mean soil moisture content from 2071 to 2100 in different SSP scenarios and that of the PiControl scenario, suggests that more regions will become drier rather than wetter in higher warming scenarios (as shown in Fig. 7 below). However, this drying trend may not be permanent or may become permanent at a later time,

according to our definition of time of emergence, compared to the wet departure. Therefore, the land area analysis detects a smaller land area with permanent dry departures than permanent wet departures.

[Figure]

Figure C7. Percentages of regions with deviation in soil moisture contents in different climate scenarios during 2071-2100 from the PiControl baseline.

In our analysis for land areas with permanent departures, the results are reported as ensemble medians. Figure C8 below shows the difference between mean and median land area with permanent departures for SSP1-2.6 and SSP3-7.0. If the results are reported in mean, the difference between land area with wet and dry departures is smaller. However, when we looked at the results individually, there could be outliers, in this case the model CNRM-ESM2-1 that reports a much higher dry departures than the others. Therefore, we think the median is a statistically more robust result and showing the ensemble median remains our choice for the main manuscript. The difference between mean and media was added to the supplement.

[Figure]

Figure C8. The differences between mean and median land area with permanent departures for SSP1-2.6.

**Minor Comments:**

6. This study analyzed 14 selected ESM, it is need to provide additional explanation of 14 selected model for brief introduction to the version and characteristics of the land model is each ESM and the number of ensemble members in part 2.1(data)

   The 14 ESMs were selected based on their completeness and the availability of the needed soil moisture content data at the beginning of our study, which was in November 2021, after which we did not change the models but did update our analyses according to changes in the data in our selected model up until June 2022.

These 14 models have the soil moisture data of all of the four SSP scenarios and piControl simulation that fits the studied timeframe, which is 2021-2100.

In Section 2.1 we stated our rationale: "We used data from all ESMs that had both the PiControl as well as all four SSPs of interest available as of 25 June 2022 (ESGF, 2022) which amounted to 14 models." To be more precise, we changed this to

"For our data ensemble, we selected all ESMs that provided simulation outputs for both the PiControl as well as all four SSPs of interest on the ESGF servers as of 1 November 2021, which amounted to 14 models. We analyzed the output data from these models as reported on the ESGF servers on 25 June 2022 (ESGF. 2022. Detailed information on these models can be found from the references provided in Table S1."

Table S1 shows the reference for each model run and the used ensemble member.

7. how about displaying the remaining 10 models in figure 2 for 80-year average values of the regional monthly total soil moisture content ffrom PiControl scenario as supplements?

We agree, and in fact 80-year average values of the regional monthly total soil moisture content from PiControl scenario are already in Figure S2 of the supplement. There was a mistake in the Figure S2 caption, it should have been "80-year average." and this is now corrected.

8. In this study, analysis results for future SSP scenarios of 14 ESM in CMIP6 were provided. It seems necessary to find out what the reliability of the results of each model scenario run is, supplementary explanations on land variable performance in historical runs, or what has been reported in previous studies.

We understand this question, but in fact this is not at all straightforward. Total soil moisture or root zone soil moisture patterns are hard to verify. Even if surface soil moisture is difficult as measurements over dense vegetation are flagged as poor quality and, moreover, the surface soil moisture patterns do not even have to coincide with root zone soil moisture. We believe that 14 models should give a reasonable estimate of the model spread and associated uncertainty.

**Response to Referee 2 Wolfgang Wagner**

With great interest I read this paper and its successor paper (Wang-Erlandsson et al. 2022) that proposed to consider the root-zone soil moisture content for determining the green water planetary boundary, namely by computing the percentage of ice-free land area on which root-zone soil moisture deviates from Holocene variability for any month of the year. I find this new concept very convincing and well justified from an Earth system perspective. Furthermore, conceptually, it is quite simple which helps in the implementation and interpretation of the results. The major uncertainty comes from the quality of the input soil moisture data sets, which unfortunately is not known for the long time periods considered. Having said that I find the scenarios computed from the multi-model ensemble (14 models from CMIP6) for the four pathways (SSP1-2.6, SSP2-4.5, SSP3-7.0, SSP5-8.5) plausible, even though I remain sceptical about the finding that there may more pronounced wetting than drying trends. E.g. can we really expect to see more soil moisture in the Sahara? Questions like these could e.g. be addressed by confronting the land surface components of the different ESMs with remotely sensed soil moisture data. But I understand that this is outside the scope of this paper. Overall, the paper is very well written and clear. Limitations are also discussed. So, in short, I recommend publishing the paper after having addressed the comments of reviewer #1.

We thank the referee for such positive comments. We would like to respond to the point of the referee remaining skeptical about finding more wetting than drying trends. Partly this may of course be caused by errors in the ESMs as the referee also seems to suggest, however, partly this may also simply be caused by the fact that:

- we looked at relative instead of absolute changes (see response to point 2, ref#1).
- General changes show more drying than permanent departures (point 5, ref#1, Fig. C7).
- Moreover, in our response to point 5, ref#1, it became apparent that the differences between wetting and drying are less pronounced when looking at the ensemble means instead of the medians (Fig. C8).

**Response to Referee 3**

The article analyses root-zone soil moisture changes under various climate scenarios. By 2050, more than 25 % of global land may experience permanent shifts in soil moisture beyond pre-industrial variability. The paper presents a solid analysis and is well written and structured. I particularly like the limitations section.

We thank the referee for the positive and constructive comments.

**Major comments:**

1. No information is provided on the model selection. Some justification is necessary on why were these 14 models selected from the around 65 models that are available in the CMIP6 archive.

   Please see our response to point 6, ref#1.

2. Many CMIP6 models respond too strongly to increasing atmospheric CO2 (i.e., they are too sensitive). The CanESM5 and UKESM1-0-LL models included in the study are two prime examples of this. EC-Earth3 and IPSL-CM6A-LR also run a little hot. Pherhaps it would be good to discuss the implications of this in the limitations section.

   This is an interesting point, which we hadn't addressed in the original submission. We added a separate bullet point about this in the limitations section:

   "ESMs respond differently to the increase in atmospheric carbon dioxide in future climate scenarios, which further propagates to their estimates of hydrological variables. The range of effective climate sensitivity (ECS) in CMIP6, 1.8 to 5.6 K (Zelinka et al., 2020), is larger than that of CMIP5, 2.1 to 4.7 K (Andrews et al., 2012), and the historical ECS range 1.5 to 4.5 K (Council, 1979). A few models which are used in our study, including CanESM5, CESM2, UKESM1, and CNRM-CM6-1, notably have ECS values exceeding the upper limit of the ECS range in CMIP5 (Zelinka et al., 2020), which likely results from stronger positive cloud feedback and the combination of weak overall negative feedback and moderate radiative forcing (Zelinka et al., 2020). The high ECS values in some models could mean more widespread and pronounced departures of soil moisture predicted in these models compared to what would be realistic."

3. The code is not available at the provided link (https://github.com/enninglai/Departure-of-soil-moisture-content-from-the-Preindus trial-Baseline). This should be resolved prior to publication of the final paper.

   We are sorry to hear that the link did not work. We have now issued a persistent identifier by uploading the code to Zenodo: https://doi.org/10.5281/zenodo.8166198 and this is updated in the code and data availability sections.

**Minor comments:**

4. Title: consider replacing "as early as 2050" with "by 2050" to make the title more concise.

   This change would make the title more concise, however, after consideration, we still prefer "as early as" as it gives more sense of urgency.

5. Figure 7: Why are some regions not shown? Is it possible to show all regions?

   We had originally included all regions, but the editor advised us to reduce the number of regions as the figure was not that clear. The full figures and tables with all regions can, however, be found in the supplement as was already indicated in the caption of Fig. 7 (now Fig. 8). We did note, however, an error in the caption, because panel (b) actually refers to all regions and this has been corrected now.

**References (that were not yet part of the manuscript)**

Andrews, T., Gregory, J. M., Webb, M. J., and Taylor, K. E.: Forcing, feedbacks and climate sensitivity in CMIP5 coupled atmosphere-ocean climate models, Geophysical Research Letters, 39, https://doi.org/10.1029/2012GL051607, 2012.

Berg, A., Sheffield, J., and Milly, P. C. D. (2017), Divergent surface and total soil moisture projections under global warming, Geophys. Res. Lett., 44, 236– 244, doi:10.1002/2016GL071921.

Council, N. R.: Carbon Dioxide and Climate: A Scientific Assessment, The National Academies Press, Washington, DC, https://doi.org/10.17226/12181, 1979.

Dai, A., Zhao, T. & Chen, J. Climate Change and Drought: a Precipitation and Evaporation Perspective. Curr Clim Change Rep 4, 301–312 (2018). https://doi.org/10.1007/s40641-018-0101-6

Hausfather, Z. and Peters, G. P.: Emissions – the 'business as usual' story is misleading, 577, 618–620, https://doi.org/10.1038/d41586-020-00177-3, 2020.

Jebeile, J., Barberousse, A. Model spread and progress in climate modelling. Euro Jnl Phil Sci 11, 66 (2021). https://doi.org/10.1007/s13194-021-00387-0

Mankin, J. S., Seager, R., Smerdon, J. E., Cook, B. I., & Williams, A. P. (2019). Mid-latitude freshwater availability reduced by projected vegetation responses to climate change. Nature Geoscience, 12(12), 983–988. https://doi.org/10.1038/s41561-019-0480-x

Reichler, T., & Kim, J. (2008). How well do coupled models simulate today's climate? Bulletin of the American Meteorological Society, 89, 303–312.

Stocker, T.,  Qin, D., Plattner G-K., Tignor M., Allen S., Boschung J., Nauels A., Xia Y., Bex V., and Midgley  P. (Eds.). (2013). IPCCFAQ 1.1: If understanding of the climate system has increased, why hasn't the range of temperature projections been reduced?. Climate change 2013: The physical science basis: summary for policymakers, technical summary and frequently asked questions. WMO, UNEP.

Zelinka, M. D., Myers, T. A., McCoy, D. T., Po-Chedley, S., Caldwell, P. M., Ceppi, P., Klein, S. A., and Taylor, K. E.: Causes of Higher Climate Sensitivity in CMIP6 Models, Geophysical Research Letters, 47, e2019GL085 782, https://doi.org/10.1029/2019GL085782, 2020.

---

## Referee Report (RR1)

Review for **"Root zone soil moisture in over 25 % of global land permanently**

**beyond pre-industrial variability as early as 2050**

The contents of this study are interesting and, in summary, this study conducted an analysis of future changes in total soil moisture across four different Shared Socioeconomic Pathway (SSP) scenarios. Utilizing data from 14 Earth System Models (ESMs) in CMIP6, the study assessed deviations in total soil moisture from the PiControl scenario, identifying permanent deviations beyond the variability observed in the PiControl scenario. The timing of emergence for these permanent deviations was also evaluated. The primary objective of this study is to systematically organize regional results in a quantitative manner, contextualized within the framework of future SSP scenarios across the global domain. In the last review, I gave major and minor comments as Ref 1. The content of the discussion was actively reflected in figures and sentences for the previous major and minor comments. Thank you for actively responding in detail to all the detailed Major& Minor comments. The discussion of some limitations is also satisfactory. Overall, it was well written, and the ability to present is evaluated as good. This study is evaluated as accept, recommending publishing the paper